# Emergency capacity analysis in Ethiopia: Results of a baseline emergency facility assessment

**Tsion Firew**[1], **Diksha Mishra**[2]*, **Tirsit Makonnen**[3], **Helena Hailu Fantaye**[4], **Bethlehem Workeye**[4], **Sofia Kebede**[5], **Fatuma Ebrahim Yimer**[6], **Yonas Abebe**[6], **Betelehem Shiferaw**[4], **Alegnta Gebreyesus**[4], **Menbeu Sultan**[6], **Aklilu Azazh**[5]

1 Columbia University Irving Medical Center, New York City, NY, United States of America, 2 Weill Cornell Medical Center, New York City, NY, United States of America, 3 Cooper University Healthcare Center, Camden, NJ, United States of America, 4 Ministry of Health Ethiopia, Addis Ababa, Ethiopia, 5 Adidas Ababa University, Addis Ababa, Ethiopia, 6 St. Paul's Hospital Millennium Medical College, Addis Ababa, Ethiopia

☯ These authors contributed equally to this work.
* dim9057@med.cornell.edu

## Abstract

### Introduction

In Ethiopia, the specialty of Emergency Medicine is a relatively new discipline. In the last few decades, policymakers have made Emergency Medicine a priority for improving population health. This study aims to contribute to this strengthening of Emergency Medicine, by conducting the country's first baseline gap analysis of Emergency Medicine Capacity at the pre-hospital and hospital level in order to help identify needs and areas for intervention.

### Methods

This is a cross sectional investigation that utilized a convenience sampling of 22 primary, general and tertiary hospitals. Trained personnel visited the hospitals and conducted 4-hour interviews with hospital administrators and emergency care area personnel. The tool used in the interview was the Columbia University sidHARTe Program Emergency Services Resource Assessment Tool (ESRAT) to evaluate both emergency and trauma capacity in different regions of Ethiopia. The findings of this survey were then compared against two established standards: the World Health Organization's Essential Package of Emergency Care (EPEC), as well as those set by Ethiopia's Federal Ministry of Health.

### Results

The tool assessed the services provided at each hospital and evaluated the infrastructure of emergency care at the facility. Triage systems differed amongst the hospitals surveyed though triaging and emergency unit infrastructures were relatively similar amongst the hospitals. There was a marked variability in the level of training, guidelines, staffing, disaster preparedness, drug availability, procedures performed, and quality assurance measures from hospital to hospital. Most regional and district hospitals did not have nurses or doctors

**Data Availability Statement:** All relevant data are within the manuscript and its Supporting Information files.

**Funding:** The authors received no specific funding for this work.

**Competing interests:** The authors have declared that no competing interests exist.

trained in Emergency Medicine and over 70% of the hospitals did not have written guidelines for standardized emergency care.

## Conclusion

This gap analysis has revealed numerous inconsistencies in health care practice, resources, and infrastructure within the scope of Emergency Medicine in Ethiopia. Major gaps were identified, and the results of this assessment were used to devise action priorities for the Ministry of Health. Much remains to be done to strengthen Emergency Medicine in Ethiopia, and numerous opportunities exist to make additional short and long-term improvements.

## Introduction

In Ethiopia, modern medicine began to develop during the reign of Menelik II (1865–1913). The Russian Red Cross mission in 1896 established the first hospital, The Russian Hospital, with the aim of treating soldiers injured during the Ethio-Italian War (the Adwa war). In 1906, the Russian Red Cross left Ethiopia and the hospital came under government control. This hospital was renamed after Menelik II [1–3]. Over the coming years, health centers were established throughout Ethiopia to provide primary outpatient and minimal inpatient services to the surrounding population. However, the majority of patients lived in rural areas with their closest health center often greater than 10 kilometers from the referral hospital and they had no access to motorized transportation. As a result, most of the population had little to no access to emergency care or appropriate diagnostic equipment [4].

In the past two decades, the Federal Ministry of Health (FMOH), the Addis Ababa City and Health Bureau, and Addis Ababa University School of Medicine have spearheaded initiatives to improve emergency care, strengthen human resources through training and education, and establish emergency care systems in the country. Concrete achievements include emergency medicine graduate programs and short-term training programs, as well as the development of prehospital ambulance programs [5]. Appropriate schooling and training paves the way for standardized emergency care systems. Standardized emergency care systems have demonstrated improved patient outcomes [6] and, therefore, have an important role in improving population health and should be a priority for local policy makers. As a result, the organizations mentioned above have recommended standards which have been established by the Ethiopian Standards Agency. However, not only are many of these endeavors still in their infancy, but none of these initiatives arose from a formal gap analysis to compare the services currently available to the resources needed to adequately serve the population.

In order to further strengthen emergency care, eight years ago the FMOH established a subdivision named the Emergency and Critical Care Directorate (ECCD) that was tasked with developing and standardizing emergency care. Despite this tremendous step forward in prioritizing emergency care, the ECCD has limited enforcement and legal mandate to ensure compliance with standards, a power which was traditionally held by the Ethiopian Food and Drug Authority. Therefore, the ECCD would benefit from a close partnership with this regulatory body.

In 2016, the ECCD built a task force to evaluate the state of emergency medicine in Ethiopia. This task force consisted of local clinicians, academicians and policy makers. The major objective of this task force was to evaluate the current emergency care capacity in major

hospitals by performing a gap analysis. According to a search of the FMOH's library of data, this investigation is the first nationwide emergency and trauma gap analysis that has been conducted and published. The information obtained from this assessment will be critical for appropriate resource allocation and strengthening of the current emergency units.

## Methods

This is a cross sectional investigation that utilized convenience sampling of twenty-two hospitals consisting of primary, general and tertiary hospitals (see Table 1). This study was part of MOH's quality improvement initiative and thus no IRB consent was obtained. See the letter attached from MOH for confirmation. Verbal consent was obtained from participants. This was to assure that all participants were kept anonymous. The main survey that was used was based off of Columbia University's sidHARTe Program Emergency Services Resource Assessment Tool (ESRAT) [7]. It was originally developed by Columbia University to assess for emergency care capacity in Rwanda and Ghana. Over a period of three weeks, an ECCD task force composed of general surgeons, neurosurgeons, and emergency physicians expanded upon ESRAT to include a trauma evaluation and called it ETSRAT-Ethiopia (See S1 Appendix). ETSRAT-Ethiopia is the main tool used for this study.

Twelve interviewers consisting of physicians and nurses from various medical fields were trained by emergency physicians on how to use the ETSRAT-Ethiopia and conduct the four-hour interview. The interviewers worked in teams of two, with each pair consisting of at least one nurse or physician that specializes in Emergency Medicine. Each pair would visit a hospital and interview hospital administrators and emergency care area personnel using the ETSRAT-Ethiopia survey mentioned above. Initially, three hospitals were piloted in the capital of Ethiopia, Addis Ababa, during the first week. A further revision was made to the tools based on feedback from the pilot assessments. Using convenience sampling, a total of twenty-two hospitals were evaluated across seven regions of Ethiopia using the further revised ETSRAT-Ethiopia tool.

**Table 1. List of hospitals assessed in the study (Sources: Ethiopian Standards Agency).**

| Primary Hospital [8] | General Hospital [9] | Tertiary Hospital [10] |
|---|---|---|
| Minimum Beds | | |
| 35+ | 50+ | 110+ |
| Minimum emergency services provided | | |
| Availability 24 hours a day, 7 days a week, 365 days a year | | |
| · Basic emergency surgical intervention | | |
| · Comprehensive emergency obstetric care | | |
| · Minimum house staff: 1 Emergency surgical officer, 1 general practitioner, 2 nurses | | |
| · Protocol for 15 common presentations | | |
| · 8+ emergency room beds starting at general level | | |
| Names | | |
| • Adare Hospital<br>• Addis Alem Hospital<br>• Houzane Hospital<br>• Sabian Hospital | • Adama Hospital<br>• Dubti Hospital<br>• Gambella Hospital<br>• Mekelle University Hospital<br>• Melka Oda Hospital<br>• Minilik Hospital<br>• Shashemene Hospital<br>• Tirunesh Beijing Hospital<br>• Yirgalem Hospital | • Asela Hospital<br>• Awassa Hospital<br>• Ayder Hospital<br>• Dilchora Hospital<br>• Felegehiwot Hospital<br>• Gonder University Hospital<br>• Hiwotfana Hospital<br>• Jimma Hospital<br>• St. Paul Hospital |

The findings were tabulated using Microsoft Excel Professional Plus 2016. One member of the pair reviewed each result from the survey and verified the recorded answers with the tabulated results. Based on these results, two reviewers calculated the proportion of hospitals where each parameter was measurable. Furthermore, the reviewers used the findings of ETSRA-T-Ethiopia to then retrospectively assess how the baseline status compared against two established standards: the World Health Organization's Essential Package of Emergency Care (EPEC), as well as the Ethiopian FMOH's standards. EPEC defines a series of essential functions that are needed for an emergency care system to function completely and efficiently, and consists of components such as medications, equipment, and procedural skills (See S1 Appendix).

## Results

### Hospital characteristics

Of the twenty-two hospitals included in the evaluation, four are primary (district level), nine general (regional level) and nine tertiary (referral level) hospitals. All of the evaluated hospitals are public hospitals, and they all provide emergency and surgical care. Furthermore, all the hospitals have on-site laboratory and pharmacy services. Out of the twenty-two hospitals surveyed, twenty-one stated that they provide obstetric care while one did not respond. In the survey, the term "Emergency Unit" (EU) was defined as a single area where any critically ill patient is treated or resuscitated. Some hospitals provide emergency care for adults, children and obstetric patients in separate units. Of note, the terms Emergency Care Area or Emergency Unit or Emergency Department describes the same area for the purpose of this evaluation. We will continue to use the term "Emergency Unit" to refer to all of the above for consistency.

Access to emergency care was varied among the hospitals. The EUs were accessible 24 hours a day across all hospitals. Meanwhile, with respect to population coverage, nearly 27% of hospitals covered catchment areas of > 5 million, while 23% covered areas of 1.5–5 million. The majority of hospitals reported EU visits in the prior two months as less than 20,000, while one reported between 20–25,000. Regarding financial access, 45% of hospitals reported use of third- party insurance by patients.

Infrastructure of emergency care was also varied among the hospitals. When asked about utilities such as the availability of running water, 36% reported that it was always available, while 59% reported that it was only sometimes available. Availability of electricity was more reliable with 64% reporting always having access, and 32% reporting sometimes having access. In addition, all of the EUs in these twenty-two hospitals had ambulance parking at the entrance with 77% reporting at least one ambulance available, and 95% of the hospitals reporting wheelchair or stretcher accessibility.

### Triage and resuscitation system

All hospitals reported triage and resuscitation systems, but the design varied greatly among them. Triage and resuscitation took place in one of three major areas: the main EU, the inpatient or outpatient ward, or in another specified triage and resuscitation area (STRA). In summary, the EU was used to triage and resuscitate all patients as reported by 14% and 9% of hospitals.

For medical-surgical patients, triage and resuscitation was performed in the EU in 91% and 86% of hospitals, respectfully. STRA was used for resuscitation of medical-surgical patients in 9% of hospitals. For pediatric patients, the use of STRAs was most commonly reported for triage and resuscitation, as reported by 45% and 50% of hospitals, respectfully. EUs were the

second most commonly utilized as reported by 32% and 41% of hospitals, with inpatient or outpatient wards reported by 14% and 5%. Like pediatric patients, for OBG patients, the use of STRAs was most commonly reported for triage and resuscitation, as reported by 48% and 59% of hospitals, respectfully. EUs were the second most commonly utilized as reported by 27% and 23%, with inpatient or outpatient wards reported by 18% and 14%.

With respect to triage administration, 73% of hospitals reported having a triage protocol. The triaging officer varied by hospitals as well, with 14% reporting the duty being fulfilled by a clinical nurse (MSc), 45% reporting a clinical nurse (BSc), and 27% by health officers. Despite the presence of the triaging officer, 41% reported that all officers were always appropriately trained, while 18% reported that only some officers were trained.

## Emergency training

In Ethiopia, there is an emergency specific post-graduate degree program for nurses in emergency and critical care. For physicians, specializations in emergency medicine, anesthesia and critical care are available, but only two teaching institutions in the country currently offer these specializations. Providers at all primary, general and tertiary facilities are not required to undergo emergency-specific training as part of initial or ongoing certification. The FMOH in collaboration with the two teaching institutions has been offering short certificate programs related to Emergency medicine that are offered to all types of medical providers (eg. MDs, mid-levels). These courses include: Emergency Triage Assessment and Treatment (*ETAT*) and National Integrated Emergency Medicine (NIEM), a five day course providing healthcare workers with basic emergency medicine training, as well as Basic Life Support (BLS), Advanced Cardiac Life support (ACLS), and Basic Emergency Care (BEC). Some of the tertiary hospitals have also taken the initiative to train providers in Focused Abdominal Sonography for Trauma (FAST) and airway courses. Of the hospitals surveyed, 45% of the hospitals had personnel trained in NIEM while 23% and 41% of the hospitals had personnel trained in ETAT and BLS respectively. Only one hospital provided airway training for all the physicians that work in the Emergency unit. More than 60% of the hospitals did not have any provider trained in airway management.

## Equipment

Figure I of the S1 Appendix shows a complete list of the availability of certain triage equipment and selected essential equipment in the Emergency Unit respectively. Of the triage equipment, blood pressure cuffs along with stethoscopes were the most available equipment at triage in 82% of the hospitals while thermometers, pulse oximeters and glucometers were available in 64%, 55% and 45% of the hospitals respectively. Functional wall clocks were the least available with only 32% of hospitals stating they had them. Of the other equipment deemed essential, all hospitals had bag-valve masks and nasal cannulas while less than 10% had functional intraosseous kits and CPAPs.

## Drugs

The data also looked at the medications availability in hospitals as listed in Figure II of the S1 Appendix. Of note, any medication that fell into the given category was labeled as available. As a result, for example, it is important to note that most hospitals, when saying yes to having benzodiazepines, had only one kind of benzodiazepine. As the data shows, the majority of hospitals had only half of the medications that the WHO has listed on their EPEC checklist. Infectious agents consistently lacking amongst all of the hospitals surveyed were oral and IV antifungals, antimalarials, anthelmintics, and antivirals though all hospitals had antibiotics of

at least one variety. Medications related to substance overdoses were also lacking including narcan and activated charcoal. Tetanus and Anti-rabies vaccines were also not available. Hypertonic saline, thrombolytics, and transfusion products such as FFP were also not present in the majority of hospitals.

## Procedures

Using the WHO's EPEC procedure checklist, the same twenty-two hospitals were surveyed to see what procedures they had the capacity to perform. As Figure II in the S1 Appendix shows, the capacity to perform said procedures varied widely throughout with most hospitals having less than 50% of the procedural skills and equipment needed to perform life-saving airway, orthopedic, trauma, and other basic emergent techniques.

## Written guidelines and quality assurance

Though there are no standardized clinical charts for the emergency department, the African Federation of Emergency Medicine has developed standardized triage and trauma sheets. Out of the twenty-two hospitals assessed, 30% of the hospitals had clinical guidelines for emergency patients but less than half of them had the clinical guideline present in the emergency care area during the time of the assessment.

Over 60% of the hospitals had some form of clinical audits for quality assurance measures.

## Disaster management

The hospitals were assessed to see if they have standardized plans for disaster management and disaster preparedness. Simultaneously, the hospitals were surveyed about the availability of an alarm system to alert other departments during a mass casualty incident. 24% of the hospitals had a plan for disaster management and 17% of the hospitals had disaster preparedness guidelines. 14% of the hospitals had an alarm system for mass casualty incidents.

## Discussion

The results of this investigation show both promising trends as well as areas for improvement regarding the access and infrastructure of EUs, emergency medicine triaging systems, training in emergency medicine, equipment and procedures in the hospitals, and disaster management.

It was encouraging to find that all the hospitals surveyed were operational 24 hours a day. However, the ratio of the population to EUs is suboptimal as the WHO recommends the physician to population ratio to be 1:1000 [11] but over a quarter of the hospitals surveys covered catchment areas of over 5 million. With respect to patient transportation, 77% of hospitals reported at least one ambulance available, which is in line with the ECCD's target objectives of increasing ambulance converge from 70% to 100% [5].

The water availability in the hospitals is an area of dire concern given the need for clean water in basic sanitation, prevention of infectious pathology, and delivering of medical care. Water access objectives have been established by the WHO through the Sustainable Development Goals (SDGs) which aims to, "by 2030, achieve universal and equitable access to safe and affordable drinking water for all" [12]. This investigation revealed that across the 22 hospitals surveyed, only 36% had reliable coverage, while 59% reported unreliable coverage. This is consistent with the countrywide coverage of 55%. Ethiopia's progress as a country thus far has been such that the proportion of the population that lacked access to water dropped from 74.4% to 45% [14] in 2015.

It is also concerning that across hospitals electricity availability was not a guarantee, making it questionable whether basic hospital actions were able to be performed at any given time. However, compared to water, the availability of electricity was more reliable with 64% reporting always having access, and 32% reporting sometimes having access. Universal access to energy is listed as a SDG with the goal by 2030 to achieve "universal access to affordable, reliable and modern energy services" [12]. According to the World Bank, Ethiopia's electricity access was 12.7% in 2000, 33.1% in 2010, and 44.9% in 2018 [13]. Based on this information, it took Ethiopia roughly twenty years to triple electricity availability. With current projects underway like the Grand Ethiopian Renaissance Dam, it is possible for the country to make great strides in coverage. If the country is able to double its coverage in ten years, then coverage can reach an astonishing 90% by 2030.

This investigation has revealed that the design of triage and resuscitation areas varies greatly among hospitals surveyed leading to questionable uniformity in delivery of appropriate and timely emergency medical care. This is despite the standards set up by the Ethiopian Standard Agency (ESA) (see Figure III in S1 Appendix). All hospitals reported availability of EUs to deliver emergency services, but EUs were often not the sole site of emergency and resuscitation activities. While nearly all hospitals reported that medical and surgical patients were triaged in the EUs, for pediatric and OBG patients, less than one third of hospitals reported triaging them in the EUs. Instead, these patients were triaged in an "area for this specific patient population". When examining the responses about resuscitation designation, though the majority of hospitals resuscitated medical-surgical patients in the EU, less than half of hospitals resuscitated pediatric and OBG patients in the EU. Similarly to the triage systems, pediatric and OBG patients were most likely to be resuscitated in the STRA.

While this may not create complications when considering inter-hospital transfers, it may very well create variations in the quality of care across hospitals. For example, providing emergency care in different areas such as in the outpatient or inpatient wards could lead to a delay in providing timely care for acutely ill patients. Several studies from low-middle income settings have shown that identifying acuity early and instituting appropriate therapeutic intervention early and rapidly reduces hospital mortality [14, 15].

Emergency training is a vital issue in Ethiopia. Even though Ethiopia is one of the few countries that has an established Emergency Medicine specialization in Africa, most regional and district hospitals do not have nurses or doctors trained in Emergency Medicine. Overall, less than 50% of the surveyed hospitals had one or more Emergency Staff members that were trained either in ETAT, BLS, or NIEM. The results also show that the better resourced referral and general hospitals had more trained individuals. Moreover, the hospitals with Emergency medicine specialized nurses or doctors were more likely to have written guidelines and quality assurance methods. It is essential, to further the practice of emergency medicine, that all hospital staff associated with the EU area be trained appropriately to assure the proper delivery of emergency medicine care.

At a primary and general level, few EUs have functional equipment for management of airway and continuous monitoring. But almost all primary and general level facilities in this evaluation had intravenous fluids, nasal cannula and bag valve masks. At a tertiary level, most EUs have functional equipment for management of airway and breathing. Only some of the tertiary level facilities have continuous monitoring. But almost all of the tertiary level facilities have intravenous fluids and vasoactive agents. The evaluation of the essential equipment for triage purposes has shown that though wall clocks, stethoscopes and blood pressure cuffs were the most widely available resources, more than 40% of the hospitals did not have thermometer, glucometer or pulse oximetry at the triage stations. None of the hospitals surveyed reported having antifungals, antimalarials, or antihelminthic thought most had at least one type of

antibiotic. Hypertonic saline, thrombolytics, and transfusion products such as FFP were also not present in the majority of hospitals, leading to concerns in their management of basic emergent conditions such as stroke, myocardial infarctions, trauma, and shock.

Some studies have shown that most district and regional hospitals in low and middle income countries (LMICs) have drugs and equipment necessary for providing basic emergency care but not for advanced care [15–17]. This does not hold true in our case with our data showing that, not only is basic equipment lacking, but also human resources, training and quality assurance methods. Though most hospitals reported having written guidelines, over 70% of the hospitals did not have written guidelines available at the time of survey. Written guidelines alone might not improve patient outcomes but they have been shown to improve patient care when introduced "through a multifaceted approach, targeting knowledge, motivation, resources and organization of care" [18, 19].

Even though disaster preparedness is an important aspect of emergency and trauma care, many facilities lack a formal disaster management plan. More than 75% of the hospitals reported lack of disaster preparedness and guidelines. Only 14% of the hospitals had an alarm system to alert other departments in cases of mass casualty incidents. It is essential that these gaps and others be further studied and tangible, targeted goals be set for improvements in emergency care delivery.

## Limitations

The sample size of twenty-two public hospitals is the main weakness of this assessment, limiting the generalizability to the whole country. These hospitals were selected by convenience sampling, and are most likely to represent the best-resourced public hospitals in the country. Furthermore, in urban settings, most of the population seeks medical care in private institutions, which were not included in this assessment. Therefore, future assessments should be expanded to include more hospitals and employ random sampling of the facilities.

In regards to procedures, the assumption was made that if the equipment was present in the hospital, then the providers knew how to implement it appropriately. We understand this may be a limitation as assessing for procedural competency was beyond the scope of this study. Also, when discussing whether medications such as IV fluids, benzodiazepines, or paralytics are present, it is important to note that whether the hospital had more than one medication within the given category was not asked.

## Conclusion

The latest research in our field has shown the incredible importance proper emergency medical care plays in improving the morbidity and mortality of a population. As a result, in the last few decades, policymakers in Ethiopia have made the development of Emergency Medicine a necessity. The first step in assuring an appropriate system is created is to do a gap analysis. Our gap analysis has revealed numerous inconsistencies in health care practice, resources, and infrastructure within the scope of Emergency Medicine in Ethiopia. Major gaps were identified, and the results of this assessment were used to devise action priorities for the Ministry of Health. Much remains to be done to strengthen Emergency Medicine in Ethiopia, and numerous opportunities exist to make additional short and long-term improvements.

## Supporting information

**S1 Appendix.**
(DOCX)

**S1 Data.**
(XLSX)

**S2 Data.**
(XLSX)

## Acknowledgments

We want to thank Dr. Rachel Moresky of sidHARTe Columbia for providing the tools to assess the hospitals.

We want to appreciate the staff at the Ministry of Health, St. Paul Millenium Hospital and Tikur Anbessa Specialized Hospital Emergency Department for their time revising and contextualizing the assessment tools.

We would also like to thank Mesfin Bekele, Mekdes Tati, and Tolesa Dida.

## Author Contributions

**Conceptualization:** Tsion Firew, Diksha Mishra, Helena Hailu Fantaye, Bethlehem Workeye, Sofia Kebede, Fatuma Ebrahim Yimer, Betelehem Shiferaw, Alegnta Gebreyesus, Menbeu Sultan, Aklilu Azazh.

**Data curation:** Tsion Firew, Tirsit Makonnen, Helena Hailu Fantaye, Bethlehem Workeye, Sofia Kebede, Fatuma Ebrahim Yimer, Yonas Abebe, Betelehem Shiferaw, Alegnta Gebreyesus, Menbeu Sultan, Aklilu Azazh.

**Formal analysis:** Tsion Firew, Diksha Mishra, Tirsit Makonnen, Bethlehem Workeye, Sofia Kebede, Fatuma Ebrahim Yimer, Yonas Abebe, Alegnta Gebreyesus, Aklilu Azazh.

**Funding acquisition:** Tsion Firew.

**Investigation:** Tsion Firew, Diksha Mishra, Helena Hailu Fantaye, Bethlehem Workeye, Sofia Kebede, Fatuma Ebrahim Yimer, Yonas Abebe, Betelehem Shiferaw, Alegnta Gebreyesus, Menbeu Sultan, Aklilu Azazh.

**Methodology:** Tsion Firew.

**Project administration:** Tsion Firew, Bethlehem Workeye, Fatuma Ebrahim Yimer, Yonas Abebe, Menbeu Sultan.

**Resources:** Tsion Firew, Sofia Kebede, Fatuma Ebrahim Yimer.

**Supervision:** Tsion Firew, Bethlehem Workeye.

**Validation:** Tsion Firew, Bethlehem Workeye, Sofia Kebede, Yonas Abebe, Alegnta Gebreyesus, Menbeu Sultan, Aklilu Azazh.

**Writing – original draft:** Tsion Firew, Diksha Mishra, Helena Hailu Fantaye, Sofia Kebede, Fatuma Ebrahim Yimer, Yonas Abebe, Betelehem Shiferaw, Alegnta Gebreyesus, Menbeu Sultan, Aklilu Azazh.

**Writing – review & editing:** Diksha Mishra, Tirsit Makonnen.

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
