## [Decision Letter · Decision Letter 0]

20 May 2021

PONE-D-21-10721

Emergency Capacity Analysis in Ethiopia: Results of the First Baseline Emergency Facility Assessment

PLOS ONE

Dear Dr. Mishra,

Thank you for submitting your manuscript to PLOS ONE. After careful consideration, we feel that it has merit but does not fully meet PLOS ONE’s publication criteria as it currently stands. Therefore, we invite you to submit a revised version of the manuscript that addresses the points raised during the review process.

We look forward to receiving your revised manuscript.

Kind regards,

Itamar Ashkenazi

Academic Editor

PLOS ONE

Journal Requirements:

4. Please ensure that you include a title page within your main document. We do appreciate that you have a title page document uploaded as a separate file, however, as per our author guidelines (http://journals.plos.org/plosone/s/submission-guidelines#loc-title-page) we do require this to be part of the manuscript file itself and not uploaded separately.

5. Please include additional information regarding the survey or questionnaire used in the study and ensure that you have provided sufficient details that others could replicate the analyses. For instance, if you developed a questionnaire as part of this study and it is not under a copyright more restrictive than CC-BY, please include a copy, in both the original language and English, as Supporting Information.

6. In your Methods section, please provide additional information about data collection. Please ensure you have provided sufficient details to replicate the analyses such as:

- the date range (month and year) of data collection

- a description of any inclusion/exclusion criteria that were applied to participant/site recruitment.

7. In the Methods, please clarify that participants provided oral consent. Please also state in the Methods:

- Why written consent could not be obtained

- How oral consent was documented

For more information, please see our guidelines for human subjects research: https://journals.plos.org/plosone/s/submission-guidelines#loc-human-subjects-research

8. To comply with PLOS ONE submission guidelines, in your Methods section, please provide additional information regarding your statistical analyses. For more information on PLOS ONE's expectations for statistical reporting, please see https://journals.plos.org/plosone/s/submission-guidelines.#loc-statistical-reporting.

9. Please list the name and version of any software package used for statistical analysis, alongside any relevant references. For more information on PLOS ONE's expectations for statistical reporting, please see https://journals.plos.org/plosone/s/submission-guidelines.#loc-statistical-reporting.

10. Please include a caption for each figure in your manuscript.

11. Please ensure that you refer to Figure 1, 2, 3 and 4 in your text as, if accepted, production will need this reference to link the reader to the figure.

12. Please include titles for your Supporting Information tables.

13. Please include a copy of Table I which you refer to in your text on page 3.

14. We note you have included a table to which you do not refer in the text of your manuscript. Please ensure that you refer to Table IIIa and IIIb in your text; if accepted, production will need this reference to link the reader to the Table.

Reviewers' comments:

Reviewer's Responses to Questions

**Comments to the Author**

1. Is the manuscript technically sound, and do the data support the conclusions?

Reviewer #1: Yes

Reviewer #2: Yes

2. Has the statistical analysis been performed appropriately and rigorously? 

Reviewer #1: Yes

Reviewer #2: Yes

3. Have the authors made all data underlying the findings in their manuscript fully available?

Reviewer #1: Yes

Reviewer #2: No

4. Is the manuscript presented in an intelligible fashion and written in standard English?

Reviewer #1: Yes

Reviewer #2: Yes

5. Review Comments to the Author

Reviewer #1: On the title need modifications.. says first baseline ... need clarification because its vague

Introduction and objective is clear

On the discussion its better to write it as paragraph rather than dividing it in theme

-make it focused and precise

- after identifying the gaps in each hospital what does authors recomend from experience or interventions from other settings

Conclusion should include all aspects of the paper introduction results and recommendations.

Overall its good and new research finding

Reviewer #2: It a very important study to improve emergency care. It is important that the study was part of an ongoing study conducted by the Ministry of Health. They have used the approriate statistical methods for the study. They have presented thier data but there are missing ones within the submission, such as table 1 mentioned in line 63 and most of all the letter from Ministry of Health mentioned in line 79/ 80. Once these are completed, my recommendation is to go ahead and publish the paper.

6. PLOS authors have the option to publish the peer review history of their article (what does this mean?). If published, this will include your full peer review and any attached files.

Reviewer #1: **Yes: **Selamawit Alemayehu

Reviewer #2: No

---

## [Author Response · Author response to Decision Letter 0]

7 Sep 2021

Dear Reviewers, 

Please see the following issues addressed:

1. Please amend your Response to Reviewers letter to include a point by point response to each of the points made by the Editor and / or Reviewers. Please follow this link for more information: https://urldefense.proofpoint.com/v2/url?u=http-3A__blogs.PLOS.org_everyone_2011_05_10_how-2Dto-2Dsubmit-2Dyour-2Drevised-2Dmanuscript_&d=DwIGaQ&c=lb62iw4YL4RFalcE2hQUQealT9-RXrryqt9KZX2qu2s&r=umxSEMkcGadXDYDQmHtEVMzzj6hKkX9DmS7wSyJSuqo&m=FMEgiVO2HMjZmZldL5La-WmvZ9v7w5ewBqZDZbnKAlc&s=EF5q610FancagswuIC4I4pKko1drjC14ITtNc99UcrM&e=

The following format has now been used. 

2. Please amend the title either on the online submission form or in your manuscript so that they are identical.

The title has been revised. 

3. We note you have included a table to which you do not refer in the text of your manuscript. Please ensure that you refer to Table IIIa and IIIb in your text; if accepted, production will need this reference to link the reader to the Table.

There is a figure III that has been referred to in line 277 – but no other tables/charts can be found that have not been referred to? Please let us know if anything else is being referred to. 

4. If possible, please upload a file showing your changes either highlighted or using track changes. This should be uploaded as a Revised Manuscript w/tracked changes, file type. Please follow this link for more information: https://urldefense.proofpoint.com/v2/url?u=http-3A__blogs.PLOS.org_everyone_2011_05_10_how-2Dto-2Dsubmit-2Dyour-2Drevised-2Dmanuscript_&d=DwIGaQ&c=lb62iw4YL4RFalcE2hQUQealT9-RXrryqt9KZX2qu2s&r=umxSEMkcGadXDYDQmHtEVMzzj6hKkX9DmS7wSyJSuqo&m=FMEgiVO2HMjZmZldL5La-WmvZ9v7w5ewBqZDZbnKAlc&s=EF5q610FancagswuIC4I4pKko1drjC14ITtNc99UcrM&e=

This file has been uploaded.

We appreciate all of the reviewer’s comments and look forward to discussing our paper further as needed. Please let us know if any further changes need to be made. 

The Authors

---

## [Editor Report · Decision Letter 1]

24 Sep 2021

Emergency Capacity Analysis in Ethiopia: Results of A Baseline Emergency Facility Assessment

PONE-D-21-10721R1

Dear Dr. Mishra,

We’re pleased to inform you that your manuscript has been judged scientifically suitable for publication and will be formally accepted for publication once it meets all outstanding technical requirements.

Kind regards,

Itamar Ashkenazi

Academic Editor

PLOS ONE

---

## [Editor Report · Acceptance letter]

12 Jan 2022

PONE-D-21-10721R1 

Emergency capacity analysis in Ethiopia: Results of a baseline emergency facility assessment 

Dear Dr. Mishra:

I'm pleased to inform you that your manuscript has been deemed suitable for publication in PLOS ONE. Congratulations! Your manuscript is now with our production department. 

Kind regards, 

on behalf of

Dr. Itamar Ashkenazi 

Academic Editor

PLOS ONE